# GRADIENT DESCENT HAPPENS IN A TINY SUBSPACE

## ABSTRACT

We show that in a variety of large-scale deep learning scenarios the gradient dynamically converges to a very small subspace after a short period of training. The subspace is spanned by a few top eigenvectors of the Hessian (equal to the number of classes in the dataset), and is mostly preserved over long periods of training. A simple argument then suggests that gradient descent may happen mostly in this subspace. We give an example of this effect in a solvable model of classification, and we comment on possible implications for optimization and learning.

## 1 INTRODUCTION

Stochastic gradient descent (SGD) (Robbins & Monro, 1951) and its variants are used to train nearly every large-scale machine learning model. Its ubiquity in deep learning is connected to the efficiency at which gradients can be computed (Rumelhart et al., 1985; 1986), though its success remains somewhat of a mystery due to the highly nonlinear and nonconvex nature of typical deep learning loss landscapes (Bottou et al., 2016). In an attempt to shed light on this question, this paper investigates the dynamics of the gradient and the Hessian matrix during SGD.

In a common deep learning scenario, models contain many more tunable parameters than training samples. In such "overparameterized" models, one expects generically that the loss landscape should have many *flat directions*: directions in parameter space in which the loss changes by very little or not at all (we will use "flat" colloquially to also mean approximately flat).[1] Intuitively, this may occur because the overparameterization leads to a large redundancy in configurations that realize the same decrease in the loss after a gradient descent update.

One local way of measuring the flatness of the loss function involves the Hessian. Small or zero eigenvalues in the spectrum of the Hessian are an indication of flat directions (Hochreiter & Schmidhuber, 1997). In Sagun et al. (2016; 2017), the spectrum of the Hessian for deep learning crossentropy losses was analyzed in depth.[2] These works showed empirically that along the optimization trajectory the spectrum separates into two components: a *bulk* component with many small eigenvalues, and a *top* component of much larger positive eigenvalues.[3]

Correspondingly, at each point in parameter space the tangent space has two orthogonal components, which we will call the *bulk subspace* and the *top subspace*. The dimension of the top subspace is $k$, the number of classes in the classification objective. This result indicates the presence of many flat directions, which is consistent with the general expectation above.

In this work we present two novel observations:

- First, the gradient of the loss during training quickly moves to lie within the top subspace of the Hessian.[4] Within this subspace the gradient seems to have no special properties; its direction appears random with respect to the eigenvector basis.

---

[1] Over parameterization suggests many directions in weight space where the loss does not change. This implies that the curvature of the loss, captured through the hessian spectrum, vanishes in these directions. In the remainder of the paper, we use the term *flat*, as is common in the literature, in a slightly broader sense to describe this curvature of the loss surface, not necessarily implying vanishing of the gradient.

[2] For other recent work on the spectrum of the Hessian as it relates to learning dynamics, see Pascanu et al. (2014); Dauphin et al. (2014); Chaudhari et al. (2016).

[3] We provide our own evidence of this in Appendix B and provide some additional commentary.

[4] This is similar to Advani & Saxe (2017), who found that a large fraction of the weights in overparameterized linear models remain untrained from their initial values (thus the gradient in those directions vanishes).

- Second, the top Hessian eigenvectors evolve nontrivially but tend not to mix with the bulk eigenvectors, even over hundreds of training steps or more. In other words, the top subspace is approximately preserved over long periods of training.

These observations are borne out across model architectures, including fully connected networks, convolutional networks, and ResNet-18, and data sets (Figures 1, 2, Table 1, Appendices C-D).

Taken all together, despite the large number of training examples and even larger number of parameters in deep-learning models, these results seem to imply that learning may happen in a tiny, slowly-evolving subspace. Indeed, consider a gradient descent step $-\eta g$ where $\eta$ is the learning rate and $g$ the gradient. The change in the loss to leading order in $\eta$ is $\delta L = -\eta \|g\|^2$. Now, let $g_{\text{top}}$ be the projection of $g$ onto the top subspace of the Hessian. If the gradient is mostly contained within this subspace, then doing gradient descent with $g_{\text{top}}$ instead of $g$ will yield a similar decrease in the loss, assuming the linear approximation is valid. Therefore, we think this may have bearing on the question of how gradient descent can traverse such a nonlinear and nonconvex landscape.

To shed light on this mechanism more directly, we also present a toy model of softmax regression trained on a mixture of Gaussians that displays all of the effects observed in the full deep-learning scenarios. This isn't meant as a definitive explanation, but rather an illustrative example in which we can understand these phenomenon directly. In this model, we can solve the gradient descent equations exactly in a limit where the Gaussians have zero variance.[5] We find that the gradient is concentrated in the top Hessian subspace, while the bulk subspace has all zero eigenvalues. We then argue and use empirical simulations to show that including a small amount of variance will not change these conclusions, even though the bulk subspace will now contain non-zero eigenvalues.

Finally, we conclude by discussing some consequences of these observations for learning and optimization, leaving the study of improving current methods based on these ideas for future work.

## 2 THE GRADIENT AND THE TOP HESSIAN SUBSPACE

In this section, we present the main empirical observations of the paper. First, the gradient lies predominantly in the smaller, top subspace. Second, in many deep learning scenarios, the top and bulk Hessian subspaces are approximately preserved over long periods of training. These properties come about quickly during training.

In general, we will consider models with $p$ parameters denoted by $\theta$ and a cross-entropy loss function $L(\theta)$. We will generally use $g(\theta) \equiv \nabla L(\theta)$ for the gradient and $H(\theta) \equiv \nabla \nabla^T L(\theta)$ for the Hessian matrix of the loss function at a point $\theta$ in parameter space. A gradient descent update with learning rate $\eta$ at step $t$ is

$$\theta^{(t+1)} = \theta^{(t)} - \eta\, g\big(\theta^{(t)}\big),\tag{1}$$

and for stochastic gradient descent we estimate the gradient using a mini-batch of examples.

### 2.1 THE GRADIENT CONCENTRATES IN THE TOP SUBSPACE

For a classification problem with $k$ classes, consider a point $\theta$ in parameter space where the Hessian spectrum decomposes into a top and a bulk subspace as discussed above.[6]

Now, let $V_{\text{top}}$ be the subspace of tangent space spanned by the top $k$ eigenvectors of the Hessian; we will call this the *top subspace*. Let $V_{\text{bulk}}$ be the orthogonal subspace. The gradient at this point can be written as a sum $g(\theta) = g_{\text{top}} + g_{\text{bulk}}$ where $g_{\text{top}}$ ($g_{\text{bulk}}$) is the orthogonal projection of $g$ onto $V_{\text{top}}$ ($V_{\text{bulk}}$). The fraction of the gradient in the top subspace is then given by

$$f_{\text{top}} \equiv \frac{\|g_{\text{top}}\|^2}{\|g\|^2}.\tag{2}$$

---

[5]Other works where the dynamics of gradient descent were analyzed directly include Fukumizu; Saxe et al. (2013); Arora et al. (2018).

[6]As we have mentioned, this decomposition was originally found in Sagun et al. (2016; 2017), and we provide additional discussion of the Hessian spectrum in Appendix B.

Figure 1 shows this fraction for common datasets and network architectures during the early stages of training. The fraction starts out small, but then quickly grows to a value close to 1, implying that there is an underlying dynamical mechanism that is driving the gradient into the top subspace.

For these experiments, training was carried out using vanilla stochastic gradient descent on a variety of realistic models and dataset combinations. However, measurements of the gradient and Hessian were evaluated using the entire training set. Additionally, all of our empirical results have been replicated in two independent implementations. (See Appendix A for further details on the numerical calculation.)

In the next subsection we provide evidence that this effect occurs in a broader range of models.

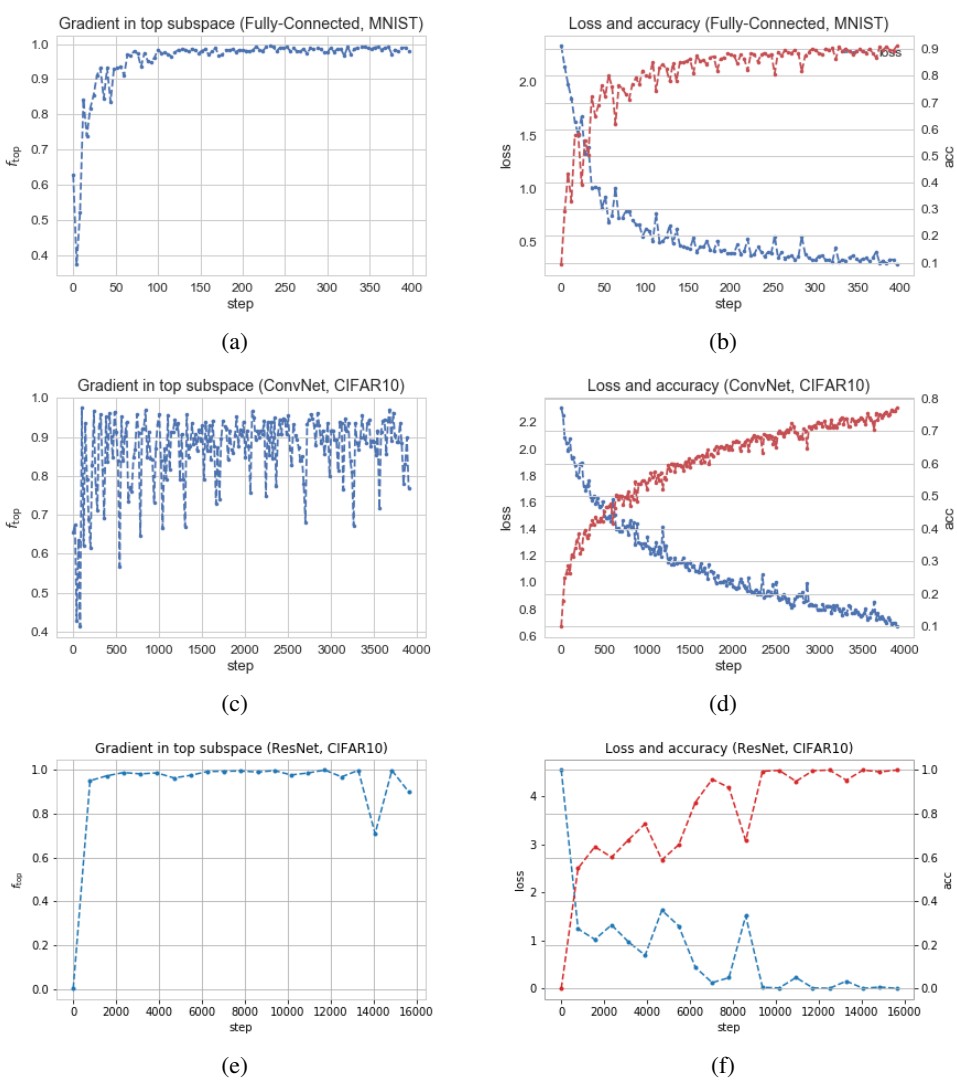

Figure 1: Fraction of the gradient in the top subspace $f_{top}$, along with training loss and accuracy. Only the initial period of training is shown, until the fraction converges. (a,b) Fully-connected network with two hidden layers with 100 neurons each, trained on MNIST using SGD with batch size 64 and $\eta = 0.1$. (c,d) Simple convolutional network (taken from Chollet et al. (2015)) trained on CIFAR10 with the same optimizer. (e,f) ResNet-18 (He et al., 2016) trained on CIFAR10.

## 2.2 HESSIAN-GRADIENT OVERLAP

In this section, we consider the overlap between the gradient $g$ and the Hessian-gradient product $Hg$ during training, defined by

$$\text{overlap}(g, Hg) \equiv \frac{g^T H g}{\|g\| \cdot \|Hg\|} .$$ (3)

The overlap takes values in the range $[-1, 1]$.

Computing the overlap is computationally much more efficient than computing the leading Hessian eigenvectors. We argue below that the overlap becomes big (of order 1) if the gradient is contained in the top subspace of the Hessian. We can use the overlap as a proxy measurement: if the overlap is large, we take that to be evidence that the gradient lives mostly in the top subspace. We measured the overlap in a range of deep learning scenarios, and the results are shown in Table 1. In these experiments we consider fully-connected networks, convolutional networks, a ResNet-18 (He et al., 2016), as well as networks with no hidden layers, models with dropout (Srivastava et al., 2014) and batch-norm (201), models with a smooth activation function (e.g. softplus instead of ReLU), models trained using different optimization algorithms (SGD and Adam), models trained using different batch sizes and learning rates, models trained on data with random labels (as was considered by Zhang et al. (2016)), and a regression task. The overlap is large for the gradient and Hessian computed on a test set as well (except for the case where the labels are randomized). In addition, we will see below that the effect is not unique to models with cross-entropy loss; a simpler version of the same effect occurs for linear and deep regression models. In all the examples that we checked, the overlap was consistently close to one after some training.

Let us now show that the overlap tends to be large for a random vector in the top Hessian subspace. Let $\lambda_i$ be the Hessian eigenvalues in the top subspace of dimension $k$, with corresponding eigenvectors $v_i$. Let $w$ be a vector in this subspace, with coefficients $w_i$ in the $v_i$ basis. To get an estimate for the overlap equation 3, we choose $w$ to be at a random vertex on the unit cube, namely choosing $w_i = \pm 1$ at random for each $i$. The overlap is then given by

$$\text{overlap}(w, Hw) = \frac{\sum_i^k \lambda_i w_i^2}{\sqrt{\left(\sum_j^k w_j^2\right)\left(\sum_l^k \lambda_l^2 w_l^2\right)}} = \frac{\sum_i^k \lambda_i}{\sqrt{k \sum_j^k \lambda_j^2}} .$$ (4)

As discussed above, in typical scenarios the spectrum will consist of $k$ positive eigenvalues where $k$ is the number of classes and all the rest close to zero. To get a concrete estimate ,we approximate this spectrum by taking $\lambda_i \propto i$ (a rough approximation, empirically, when $k = 10$), and take $k$ large so that we can compute the sums approximately. This estimate for the overlap is $\sqrt{3/4} \approx 0.87$, which is in line with our empirical observations. This should compared with a generic random vector not restricted to the top subspace, which would have an overlap much less than 1.

We have verified empirically that a random unit vector $w$ in the top Hessian subspace will have a large overlap with $Hw$, comparable to that of the gradient, while a random unit vector in the full parameter space has negligible overlap. Based on these observations, we will take the overlap equation 3 to be a proxy measurement for the part of the gradient that lives in the top Hessian subspace.

## 2.3 EVOLUTION OF THE TOP SUBSPACE

We now show empirically that the top Hessian subspace is approximately preserved during training. Let the top subspace $V_{\text{top}}^{(t)}$ at training step $t$ be spanned by the top $k$ Hessian eigenvectors $v_1^{(t)}, \ldots, v_k^{(t)}$. Let $P_{\text{top}}^{(t)}$ be the orthogonal projector onto $V_{\text{top}}^{(t)}$, defined such that $\left(P_{\text{top}}^{(t)}\right)^2 = P_{\text{top}}^{(t)}$. We will define the overlap between a subspace $V_{\text{top}}^{(t)}$ and a subspace $V_{\text{top}}^{(t')}$ at a later step $t' > t$ as follows.

$$\text{overlap}\left(V_{\text{top}}^{(t)}, V_{\text{top}}^{(t')}\right) \equiv \frac{\text{Tr}\left(P_{\text{top}}^{(t)} P_{\text{top}}^{(t')}\right)}{\sqrt{\text{Tr}\left(P_{\text{top}}^{(t)}\right)\text{Tr}\left(P_{\text{top}}^{(t')}\right)}} = \frac{1}{k} \sum_{i=1}^{k} \left\|P_{\text{top}}^{(t)} v_i^{(t')}\right\|^2 .$$ (5)

Table 1: Mean overlap results for various cases. FC refers to a fully-connected network with two hidden layers of 100 neurons each and ReLU activations. ConvNet refers to a convolutional network taken from Chollet et al. (2015). By default, no regularization was used. The regression data set was sampled from one period of a sine function with Gaussian noise of standard deviation $0.1$. We used SGD with a mini-batch size of 64 and $\eta = 0.1$, unless otherwise specified. All models were trained for a few epochs, and the reported overlap is the mean over the last 1,000 steps of training. Plots of $f_{\text{top}}$ for many of these experiments are collected in Appendix D.

| DATASET | MODEL | COMMENT | MEAN OVERLAP |
|---|---|---|---|
| MNIST | Softmax | | 0.96 |
| MNIST | FC | Softplus activation | 0.96 |
| MNIST | FC | $\eta = 0.01$ | 0.96 |
| MNIST | FC | Batch size 256 | 0.97 |
| MNIST | FC | Random labels | 0.86 |
| CIFAR10 | ConvNet | Random labels | 0.86 |
| CIFAR10 | ConvNet | Dropout, batch-norm, and extra dense layer | 0.93 |
| CIFAR10 | ConvNet | Optimized using Adam | 0.89 |
| Regression | FC | Batch size 100 | 0.99 |

It is easy to verify the rightmost equality. In particular, each element in the sum measures the fraction of a late vector $v_i^{(t')}$ that belongs to the early subspace $V_{\text{top}}^{(t)}$. Notice that the overlap of a subspace with itself is 1, while the overlap of two orthogonal subspaces vanishes. Therefore, this overlap is a good measure of how much the top subspace changes during training.[7]

Figure 2 shows the evolution of the subspace overlap for different starting times $t_1$ and future times $t_2$, and for classification tasks with $k = 10$ classes. For the subspace spanned by the top $k$ eigenvectors we see that after about $t_1 = 100$ steps the overlap remains significant even when $t_2 - t_1 \gg t_1$, implying that the top subspace does not evolve much after a short period of training. By contrast, the subspace spanned by the next $k$ eigenvectors does not have this property: Even for large $t_1$ the subspace overlap decays quickly in $t_2$.

This means that the projector $P_{\text{top}}^{(t)}$ is only weakly dependent on time, making the notion of a "top subspace" approximately well-defined during the course of training. It is this observation, in conjunction with the observation that the gradient concentrates in this subspace at each point along the trajectory, that gives credence to the idea that gradient descent happens in a tiny subspace.[8]

In Appendix C we give additional results on the evolution of the top subspace, by studying different sizes of the subspace. To summarize this, we can average the overlap over different interval values $t_2 - t_1$ for each fixed $t_1$ and plot as a function of subspace dimension. We present this plot in Figure 3 for the same fully-connected (a) and ResNet-18 (b) models as in Figure 1. Here, we very clearly see that increasing the subspace until $d = 9$ leads to a pretty fixed overlap as a function of dimension. At $d = 10$ it begins to decrease monotonically with increasing dimension. This is strong evidence that there's and interesting feature when the dimension is equal to the number of classes.[9]

## 3 A TOY MODEL

In order to understand the mechanism behind the effects presented in the previous section, in this section we work out a toy example. We find this to be a useful model as it captures all of the effects

---

[7] We have written the middle expression in (equation 5) to make it clear that our overlap is the natural normalized inner product between the projectors $P_{\text{top}}^{(t)}$ and $P_{\text{top}}^{(t')}$. This is simply related to the Frobenius norm of the difference between the two projectors, $||P_{\text{top}}^{(t)} - P_{\text{top}}^{(t')}||$, the canonical distance between linear subspaces.

[8] Note that this does not mean the actual top eigenvectors are similarly well-defined, indeed we observe that sometimes the individual eigenvectors within the subspace tend to rotate quickly and other times they seem somewhat fixed.

[9] It might be more reasonable to describe this transition at the number of classes minus one, $k - 1$, rather than the number of classes $k$. This distinction is inconclusive given the spectrum (see Appendix B), but seems rather sharp in Figure 3.

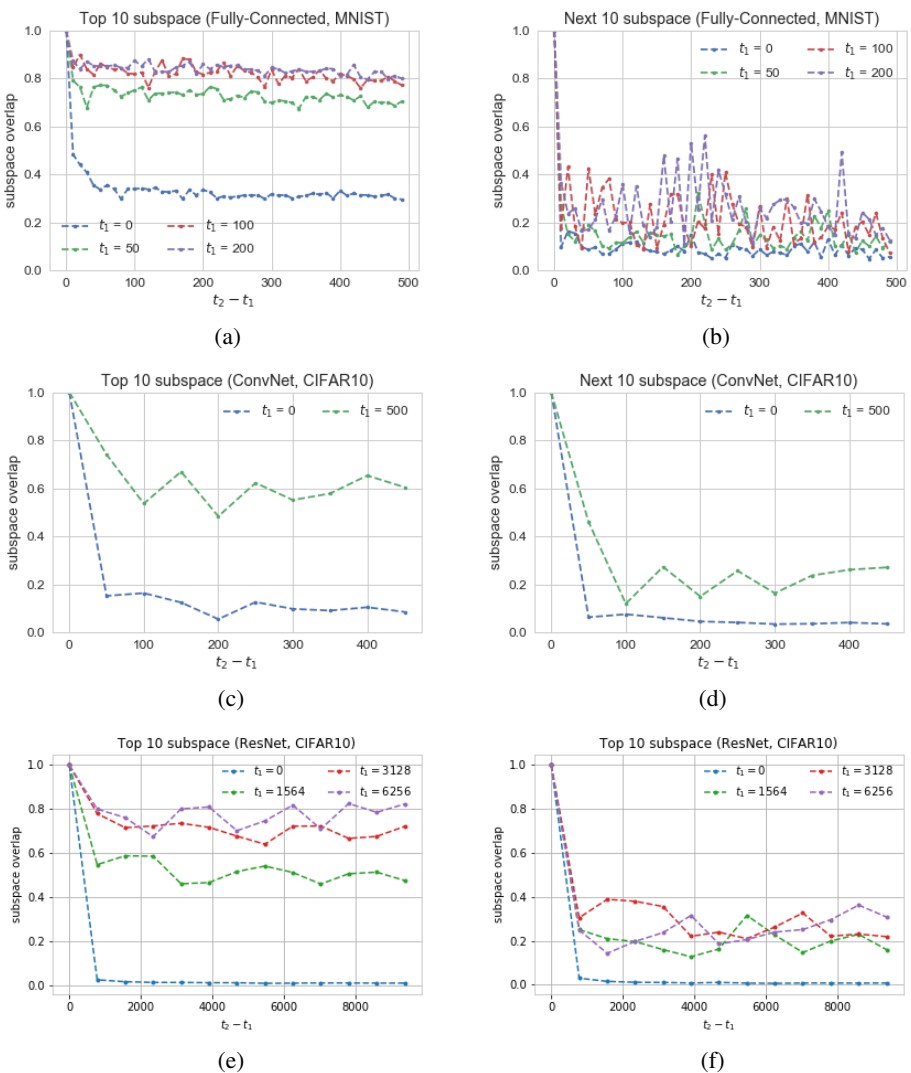

Figure 2: Overlap of top Hessian subspaces $V_{\text{top}}^{(t_1)}$ and $V_{\text{top}}^{(t_2)}$. (a) Top 10 subspace of fully-connected network trained on MNIST. (b) Subspace spanned by the next 10 Hessian eigenvectors. (c) Top 10 subspace of convolutional network trained on CIFAR10. (d) Subspace spanned by the next 10 Hessian eigenvectors. (e) Top 10 subspace of ResNet-18 trained on CIFAR10. (f) Subspace spanned by the next 10 Hessian eigenvectors. The network architectures are the same as in Figure 1.

we observed in realistic deep learning examples. However, at this point we only interpret the toy model to be illustrative and not a definitive explanation of the phenomenon.[10]

Although the way we first set it up will be very simple, we can use it as a good starting point for doing small perturbations and generalizations in which all of the realistic features are present. We will show empirically that such small perturbations do not change the qualitative results, and leave an analytic study of this perturbation theory and further generalization to future work.

Consider the following 2-class classification problem with $n$ samples $\{(x_a, y_a)\}_{a=1}^n$ with $x_a \in \mathbb{R}^d$ and labels $y_a$. The samples $x_a$ are chosen from a mixture of two Gaussian distributions $\mathcal{N}(\mu_1, \sigma^2)$ and $\mathcal{N}(\mu_2, \sigma^2)$, corresponding to the two classes. The means $\mu_{1,2}$ are random unit vectors. On this data we train a model of softmax-regression, with parameters $\theta_{y,i}$ where $y = 1, 2$ is the label and

---

[10]It is also useful in understanding how results might change as hyperparameters, e.g. the learning rate, are varied.)

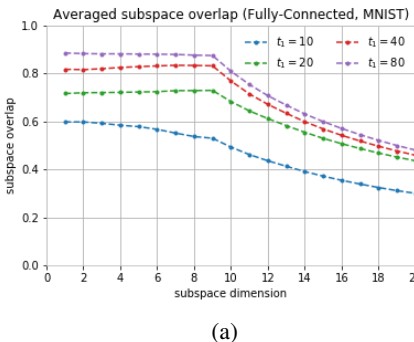 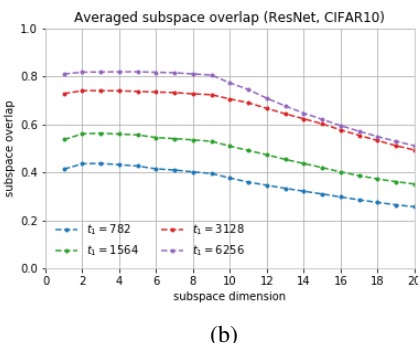

(a)                                           (b)

Figure 3: Subspace overlap of top Hessian subspaces $V_{\text{top}}^{(t_1)}$ and $V_{\text{top}}^{(t_2)}$ for different top subspace dimensions with different initial number of steps $t_1$ averaged over the interval $t_2 - t_1$ for (a) fully-connected two-layer network trained on MNIST and (b) ResNet-18 architecture trained on CIFAR10. Note the kink around subspace dimension equal to one less than the number of classes in the dataset.

$i = 1, \ldots, d$. The cross-entropy loss is given by

$$L(\theta) = -\frac{1}{n} \sum_{a=1}^{n} \log \left( \frac{e^{\theta_{y_a} \cdot x_a}}{\sum_y e^{\theta_y \cdot x_a}} \right) . \tag{6}$$

(Here we denote by $\theta_y \in \mathbb{R}^d$ the weights that feed into the $y$ logit.) We will now make several simplifying approximations. First, we take the limit $\sigma^2 \to 0$ such that the samples concentrate at $\mu_1$ and $\mu_2$. The problem then reduces to a 2-sample learning problem. Later on we will turn on a small $\sigma^2$ and show that our qualitative results are not affected. Second, we will assume that $\mu_1$ and $\mu_2$ are orthogonal. Random vectors on the unit sphere $S^{d-1}$ have overlap $d^{-1/2}$ in expectation, so this will be a good approximation at large $d$.

With these assumptions, it is easy to see that the loss function has $2d - 2$ flat directions. Therefore the Hessian has rank 2, its two nontrivial eigenvectors are the top subspace, and its kernel is the bulk subspace. The gradient is always contained within the top subspace.

In Appendix E, we use these assumptions to solve analytically for the optimization trajectory. At late-times in a continuous-time approximation, the solution is

$$\theta_{1,2}(t) = \tilde{\theta}_{1,2} + \tilde{\theta}' \pm \frac{\mu_1}{2} \log \left( \eta t + c_1 \right) \mp \frac{\mu_2}{2} \log \left( \eta t + c_2 \right), \tag{7}$$

$$g_{\theta_1}(t) = \frac{2(\mu_2 - \mu_1)}{\eta t} + \mathcal{O}(t^{-2}), \qquad g_{\theta_1}(t) = -g_{\theta_2}(t), \tag{8}$$

$$H(t) = \frac{1}{2\eta t} \begin{pmatrix} +1 & -1 \\ -1 & +1 \end{pmatrix} \otimes \left[ \mu_1 \mu_1^T + \mu_2 \mu_2^T \right] + \mathcal{O}(t^{-2}). \tag{9}$$

Here $\eta$ is the learning rate, $c_i$ are arbitrary positive real numbers, $\tilde{\theta}_i \in \mathbb{R}^d$ are two arbitrary vectors orthogonal to both $\mu_{1,2}$, and $\tilde{\theta}' \in \mathbb{R}^d$ is an arbitrary vector in the space spanned by $\mu_{1,2}$.[11] Together, $c_i$, $\tilde{\theta}_i$, and $\tilde{\theta}'$ parameterize the $2d$-dimensional space of solutions. This structure implies the following.

1. The Hessian has two positive eigenvalues (the top subspace),[12] while the rest vanish. The top subspace is always preserved.

2. The gradient evolves during training but is always contained within the top subspace.

---

[11] We thank Vladimir Kirilin for pointing out a mistake in an earlier version of this paper.

[12] For the analytically simple form of model chosen here, the two eigenvalues in this top subspace are equal. However, this degeneracy can be broken in a number of ways such as adding a bias.

These properties are of course obvious from the counting of flat directions above. We have verified empirically that the following statements hold as well.[13]

- If we introduce small sample noise (*i.e.* set $\sigma^2$ to a small positive value), then the bulk of the Hessian spectrum will contain small non-zero eigenvalues (suppressed by $\sigma^2$), and the gradient will still evolve into the top subspace.

- If we add biases to our model parameters, then the degeneracy in the top subspace will be broken. During training, the gradient will become aligned with the eigenvector that has the *smaller* of the two eigenvalues.

- All these statements generalize to the case of a Gaussian mixture with $k > 2$ classes.[14] The top Hessian subspace will consist of $k$ positive eigenvalues. If the degeneracy is broken by including biases, there will be $k-1$ large eigenvalues and one smaller (positive) eigenvalue, with which the gradient will become aligned.

### 3.1 Mostly Preserved Subspace, Evolving Gradient

Let us now tie these statements into a coherent picture explaining the evolution of the gradient and the Hessian.

The dynamics of the gradient within the top subspace (and specifically that fact that it aligns with the minimal eigenvector in that subspace) can be understood by the following argument. Under a single gradient descent step, the gradient evolves as

$$g^{(t+1)} = g\left(\theta^{(t)} - \eta g^{(t)}\right) = \left(1 - \eta H^{(t)}\right)g^{(t)} + \mathcal{O}(\eta^2). \tag{10}$$

If we assume the linear approximation holds, then for small enough $\eta$ this evolution will drive the gradient toward the eigenvector of $H$ that has the minimal, non-zero, eigenvalue. This seems to explain why the gradient becomes aligned with the smaller of the two eigenvectors in the top subspace when the degeneracy is broken. (It is not clear that this explanation holds at late times, where higher order terms in $\eta$ may become important.)[15]

The reader may wonder why the same argument does not apply to the yet smaller (or vanishing) eigenvalues of the Hessian that are outside the top subspace. Applying the argument naively to the whole Hessian spectrum would lead to the erroneous conclusion that the gradient should in fact evolve into the bulk. Indeed, from equation 10 it may seem that the gradient is driven toward the eigenvectors of $(1 - \eta H)$ with the largest eigenvalues, and these span the bulk subspace of $H$.

There are two ways to see why this argument fails when applied to the whole parameter space. First, the bulk of the Hessian spectrum corresponds to exactly flat directions, and so the gradient vanishes in these directions. In other words, the loss function has a symmetry under translations in parameter space, which implies that no dynamical mechanism can drive the gradient toward those tangent vectors that point in flat directions. Second, in order to show that the gradient converges to the bulk we would have to trust the linear approximation to late times, but (as mentioned above) there is no reason to assume that higher-order corrections do not become large.

#### Adding sample noise

Let us now discuss what happens when we introduce sample noise, setting $\sigma^2$ to a small positive value. Now, instead of two samples we have two sets of samples, each of size $n/2$, concentrated around $\mu_1$ and $\mu_2$. We expect that the change to the optimization trajectory will be small (namely

---

[13] In our experiments we used $d = 1000$, $k = 2, 5, 10$, and $\sigma = 0, 0.02$. For the means $\mu_i$, we use random unit vectors that are not constrained to be orthogonal.

[14] This can be studied analytically and will be presented in future work (Kirilin et al.). However, we will discuss an important point here of the $k > 2$ class model that makes the dynamical nature of the top-$k$ subspace more apparent. Considering the loss equation 6 and $k$ orthogonal mean vectors, one can see that symmetries of the loss lead to $k(k-1)$ nontrivial directions, meaning the Hessian is naturally rank $k(k-1)$. After solving the model, one can see that in fact this $k(k-1)$ subspace dynamically becomes dominated by $k$ top eigenvalues.

[15] We mention in passing that the mechanism above holds exactly for linear regression with quadratic loss. In this setting the Hessian is constant and there are no higher-order corrections, and so the gradient will converge to the leading eigenvector of $(1 - \eta H)$.

suppressed by $\sigma^2$) because the loss function is convex, and because the change to the optimal solution is also suppressed by $\sigma^2$. The noise breaks some of the translation symmetry of the loss function, leading to fewer flat directions and to more non-zero eigenvalues in the Hessian, appearing in the bulk of the spectrum. The Hessian spectrum then resembles more closely the spectra we find in realistic examples (although the eigenvalues comprising the top subspace have a different structure). Empirically we find that the top subspace still has two large eigenvalues, and that the gradient evolves into this subspace as before. Therefore turning on noise can be treated as a small perturbation which does not alter our analytic conclusions. We leave an analytic analysis of the problem including sample noise to future work. We note that the argument involving equation 10 can again not be applied to the whole parameter space, for the same reason as before. Therefore, there is no contradiction between that equation and saying that the gradient concentrates in the top subspace.

## 4 DISCUSSION

We have seen that quite generally across architectures, training methods, and tasks, that during the course of training the Hessian splits into two slowly varying subspaces, and that the gradient lives in the subspace spanned by the $k$ eigenvectors with largest eigenvalues (where $k$ is the number of classes). The fact that learning appears to concentrate in such a small subspace with all positive Hessian eigenvalues might be a partial explanation for why deep networks train so well despite having a nonconvex loss function. The gradient essentially lives in a convex subspace, and perhaps that lets one extend the associated guarantees to regimes in which they otherwise wouldn't apply.

An essential question of future study concerns further investigation of the nature of this nearly preserved subspace. From Section 3, we understand, at least in certain examples, why the spectrum splits into two blocks as was first discovered by Sagun et al. (2016; 2017). However, we would like to further understand the hierarchy of the eigenvalues in the top subspace and how the top subspace mixes with itself in deep learning examples. We'd also like to investigate more directly the different eigenvectors in this subspace and see whether they have any transparent meaning, with an eye towards possible relevance for feature extraction.

Central to our claim about learning happening in the top subspace was the fact the decrease in the loss was predominantly due to the projection of the gradient onto this subspace. Of course, one could explicitly make this projection onto $g_{\text{top}}$ and use that to update the parameters. By the argument given in the introduction, the loss on the current iteration will decrease by almost the same amount if the linear approximation holds. However, updating with $g_{\text{top}}$ has a nonlinear effect on the dynamics and may, for example, alter the spectrum or cause the top subspace to unfreeze. Further study of this is warranted.

Similarly, given the nontrivial relationship between the Hessian and the gradient, a natural question is whether there are any practical applications for second-order optimization methods (see Bottou et al. (2016) or Dennis Jr & Schnabel (1996) for a review). Much of this will be the subject of future research, but we will conclude by making a few preliminary comments here.

An obvious place to start is with Newton's method (Dennis Jr & Schnabel, 1996). Newton's method consists of the parameter update $\theta^{(t+1)} = \theta^{(t)} - H^{-1}g^{(t)}$. There are a few traditional criticisms of Newton's method. The most practical is that for models as large as typical deep networks, computation of the inverse of the highly-singular Hessian acting on the gradient is infeasible. Even if one could represent the matrix, the fact that the Hessian is so ill-conditioned makes inverting it not well-defined. A second criticism of Newton's method is that it does not strictly descend, but rather moves towards critical points, whether they are minima, maxima, or saddles (Pascanu et al., 2014; Dauphin et al., 2014). These objections have apparent simple resolutions given our results. Since the gradient predominantly lives in a tiny nearly-fixed top subspace, this suggests a natural low rank approximation to Newton's method

$$\theta^{(t+1)} = \theta^{(t)} - (H_{\text{top}}^{(t)})^{-1}g_{\text{top}}^{(t)}. \tag{11}$$

Inverting the Hessian in the top subspace is well-defined and computationally simple. Furthermore, the top subspace of the Hessian has strictly positive eigenvalues, indicating that this approximation to Newton's method will descend rather then climb. Of course, Newton's method is not the only second-order path towards optima, and similar statements apply to other methods.

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

## A    NUMERICAL METHODS

For the empirical results in this paper, we did not actually have to ever represent the Hessian. For example, to compute the top eigenvectors of the Hessian efficiently, we used the Lanczos method (Lehoucq et al., 1998), which relies on repeatedly computing the Hessian-vector product $Hv$ for some vector $v$. This product can be computed in common autograd packages such as TensorFlow (Abadi et al.) or PyTorch (Paszke et al., 2017) as follows. Let $v$ be a pre-computed numerical vector (such as the gradient). One first computes the scalar $a = \nabla L^T v$, and then takes the gradient of this expression, resulting in $\nabla a = Hv$.

## B    HESSIAN SPECTRUM

As first explored by Sagun et al. (2016; 2017), the Hessian eigenvalue spectrum appears to naturally separate into "top" and "bulk" components, with the top consisting of the largest $k$ eigenvalues, and the bulk consisting of the rest.

An example of this for a small fully-connected two-layer network is shown in Figure 4. The hidden layers each have 32 neurons, and the network was trained on MNIST for 40 epochs. The eigenvalues belonging to the top subspace are clearly visible, and for clarity, we labeled them showing that there's 10 nontrivial eigenvalues. We further confirmed this effect by studying datasets with a different number of classes (such as CIFAR100) and by studying synthetic datasets.

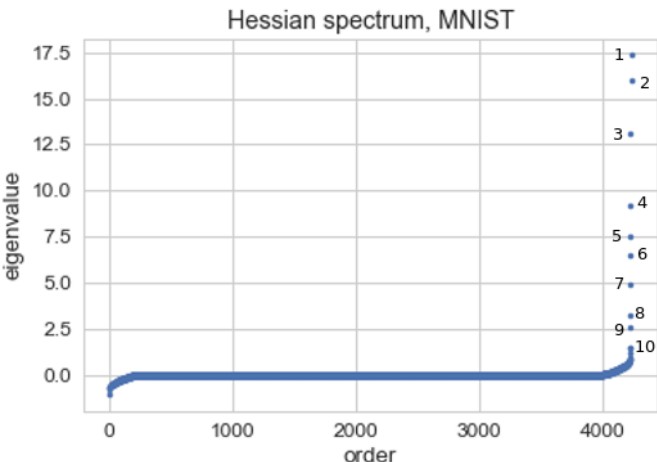

Figure 4: Eigenvalues of the Hessian of a fully-connected network with two hidden layers, each with 32 neurons, trained on MNIST for 40 epochs. The top 10 largest eigenvalues are labeled and clearly form a nontrivial tail at the right edge of the spectrum.

We also confirmed that the dimension of the top subspace is tied to the classification task and not intrinsic to the dataset. For instance, we can study MNIST where we artificially label the digits according to whether they are even or odd, creating 2 class labels (even though the data intrinsically contains 10 clusters). In this case, there were only 2 large eigenvalues, signifying that the top is 2-dimensional and not 10-dimensional. Additionally, we experimented by applying a random permutation to the MNIST labels. This removed the correlation between the input and the labels, but the network could still get very high training accuracy as in Zhang et al. (2016). In this case, we still find 10 large eigenvalues.

The fact that the top subspace is frozen (as we show in Figure 2), suggests that there could be some kind of a special feature in the Hessian spectrum. To study this, we looked at a two-layer fully-connected network on CIFAR100, with each hidden layer having 256 neurons each. We chose CIFAR100 to allow us a larger value of $k$ to perhaps see something meaningful in the transition between the bulk and top subspaces. Furthermore, rather than just plotting the value of the eigenvalues as a function of their index, we made a density plot averaged over 200 realizations. This is shown

in Figure 5, where we note that the $x$-axis is log of the eigenvalue. Since we were only interested in the transition from top to bulk, we only computed the top 1000 eigenvalues. This allowed us to study a larger model (256, 256) than we did for the plot of the full spectrum in Figure 4.

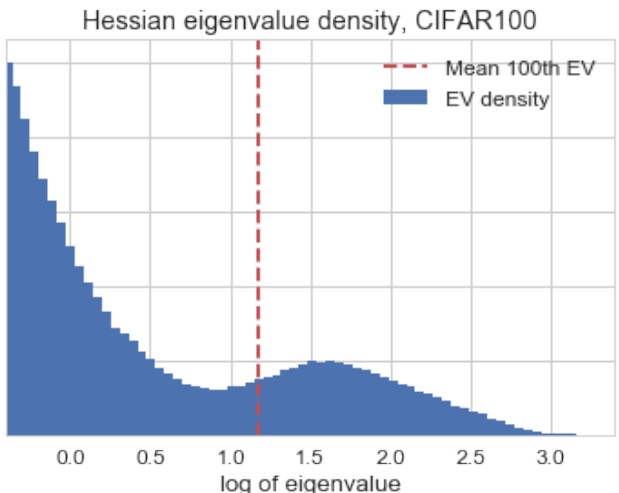

Figure 5: Histogram of eigenvalue density on the right edge of the Hessian spectrum for a fully-connected two-layer (256, 256) model trained on CIFAR100 averaged over 200 realizations.

The density plot, Figure 5, shows a clear feature in the density function describing the Hessian eigenvalues occurring around the mean 100th eigenvalue. While the exact location is hard to determine, there is a clear underdensity around the 100th eigenvalue, counting from the right edge. It's an interesting observation that a Gaussian provides a very good fit to the part of the spectrum in the top subspace, suggesting the eigenvalue distribution could be described by a log-normal distribution. However, this is only suggestive, and much more evidence and explanation is needed. In future work, it would be interesting to characterize the different functions that describe the spectral density of the Hessian.

Next, let's look at a particular top eigenvector. One hypothesis is that the corresponding eigenvectors to the $k$ largest eigenvalues would just correspond to either the weights or biases in the last layer (which also depend on the number of classes). In Figure 6, we plot the maximal eigenvector after (a) 0 steps, (b) 100 steps, (c) 200 steps, and (d) 400 steps of training for the fully-connected (100,100) architecture trained on MNIST. First it's easy to see that this vector is not constant during training. More importantly, we see that there are many nonzero elements of the vectors across the entire range of model parameters. We colored these plots according to where the parameters are located in the network, and we note that even though the top layer weights seem to have the largest coefficients, they are only $\sim 4\times$ larger than typical coefficients in the first hidden layer.

In Figure 7, we zoom in on the final layer for the fully-connected (100,100) architecture trained on MNIST after (a) 0 steps and (b) 400 steps. This makes it clear that the eigenvector is never sparse and is evolving in time. Thus, we conclude that eigenvectors are a nontrivial linear combination of parameters with different coefficients. It would be interesting to understand in more detail whether the linear combinations of parameters represented by these top-subspace eigenvectors are capturing something important about either learning dynamics or feature representation.

Finally, for completeness let us also give a plot of some example evolutions of a top Hessian eigenvalue. In Figure 8, we plot the evolution of the maximal eigenvalue for (a) our fully-connected $(100, 100)$ architecture trained on MNIST and (b) our ResNet-18 architecture trained on CIFAR10. In both cases, we see an initial period of growth, then the eigenvalue remains very large as the model is training, then it decays. The fully-connected MNIST example trains very quickly, but comparing with Figure 1 (f) for the ResNet-18, we see that the loss and accuracy converge around step 10000, where the maximum eigenvalue begins to oscillate and also decay. Our toy model suggests that eigenvalues should decay at the late part of training like $\sim 1/t$. These plots are too rough to say

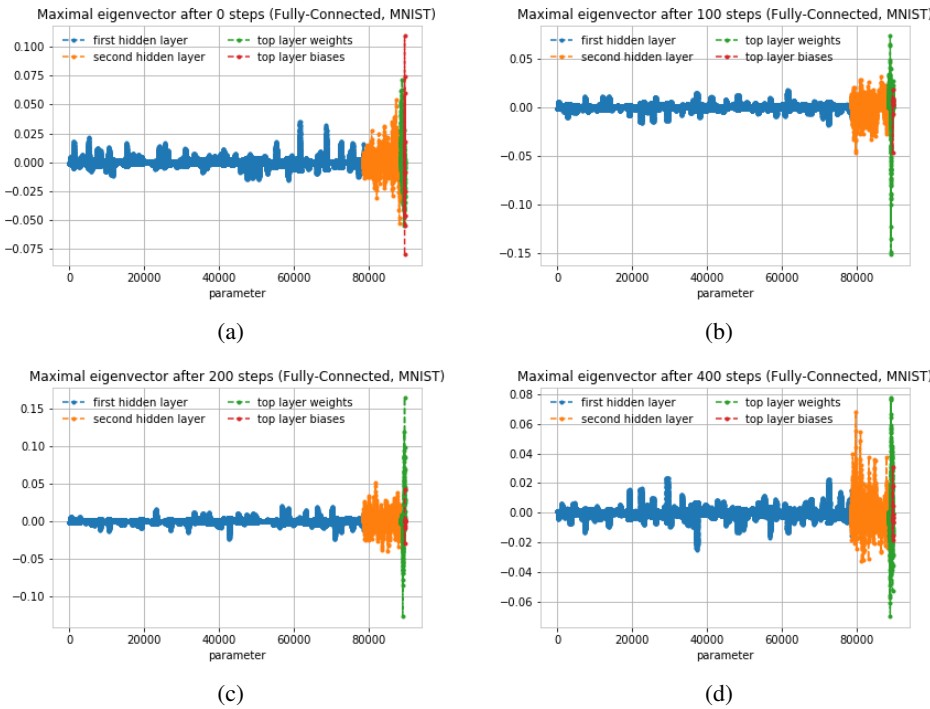

Figure 6: Eigenvector corresponding to the maximal eigenvalue for the fully-connected (100,100) architecture trained on MNIST after (a) 0 steps, (b) 100 steps, (c) 200 steps, and (d) 400 steps. We organize according to first hidden layer (blue), second hidden layer (orange), top layer weights (green), and top layer biases (red).

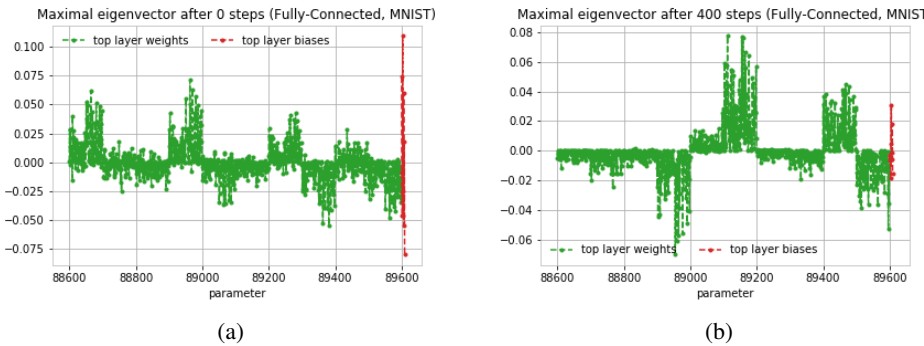

Figure 7: Eigenvector corresponding to the maximal eigenvalue for the fully-connected (100,100) architecture trained on MNIST after (a) 0 steps and (b) 400 steps zoomed in on the top layer weights and biases. These plots are strong evidence that eigenvector is clearly not dominated by any particular parameter and is meaningfully changing in time.

anything specific about the functional form of the decay, but we do see qualitatively in both cases that it's decreasing.[16]

---

[16]To learn something more concrete, ideally we should train a large number of realizations and then average the behavior of the maximal eigenvalue across the different runs. We will save this analysis for the future.

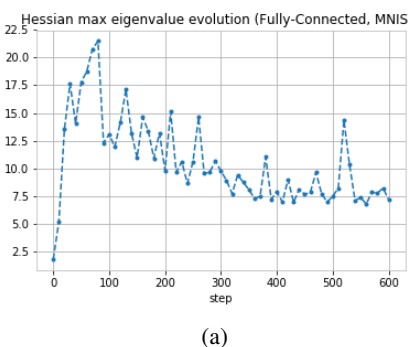 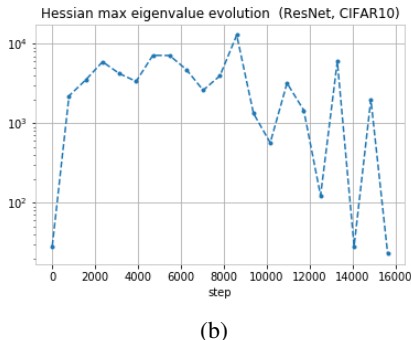

(a)                                    (b)

Figure 8: Evolution of the maximal eigenvalue for (a) fully-connected (100,100) architecture trained on MNIST and (b) ResNet-18 architecture trained on CIFAR10. Note the second plot has a log scale on the $y$-axis.

## C  $k$ IS FOR CLASSES

In this section, we will give further evidence that the size of the nearly-preserved subspace is related to the number of classes. As we showed in the last section and Figure 5 in particular, there is a feature in the Hessian spectrum that seems related to the number of classes. In Figure 1, we explain that the gradient tends to lie in a subspace spanned by the eigenvalues corresponding to the top-$k$ eigenvectors, and in Figure 2, we show that a subspace of size $k$ seems to be nearly preserved over the course of training. These three phenomena seem to be related, and here we'd like to provide more evidence.

First, let's investigate whether the nearly preserved subspace is $k$-dimensional. To do so, let us consider the same fully-connected two-layer network considered in (a) and (b) of Figure 2. In Figure 9, we consider top subspaces of different dimensions, ranging from 2 to 20. We can consider subspace dimensions of different sizes for the ResNet-18 architecture considered in (e) and (f) of Figure 2, which also has 10 classes. These results are shown in Figure 10. Both of these results show interesting behavior as we increase the subspace past the number of classes.

Notably, the top 15 and top 20 subspaces shown in (e) and (f) of Figures 9-10 and are significantly less preserved than the others. The top 11 subspace is marginally less preserved, and most of the subspaces with dimensions less than 10 seem to be preserved amongst themselves. In particular, both (e) and (f) in both plots shows that adding additional eigenvectors does not always lead to increased preservation. The maximally (i.e. largest dimensional) preserved subspace seems to peak around the number of classes. The fact that these smaller top subspaces are also preserved suggests additional structure perhaps related to the eigenvectors no longer rotating as much amongst themselves as training progresses. A nice summary of these results where we average the overlap for a particular $t_1$ over the interval $t_2 - t_1$ is shown in the main text in Figure 3.

Now that we've studied whether the fixed subspace is really $k$-dimensional, let's better understand how the fraction of the gradient spreads across the top subspace for a few different points in training. Let us define the overlap of the gradient with a particular eigenvector

$$c_i^2 \equiv \frac{\|v_i \cdot g\|^2}{\|g\|^2}, \tag{12}$$

where the numerator represents the overlap of the $i$th eigenvector (order from eigenvectors corresponding to the largest eigenvalues to the least), and the numerator is the norm squared of the $i$th overlap. This satisfies $\sum c_i^2 = 1$ when summed overall all $p$ parameter directions.

In Figure 11, we plot $c_i^2$ for (a) 0 steps (b) 50 steps (c) 100 steps, and (d) 200 steps of training for the $i$ corresponding to the top and next subspace ($i = 1 \ldots 20$) for the fully-connected (100,100) network trained on MNIST. Importantly, these plots make it clear that the gradient is not simply an eigenvector of the Hessian. In particular, before any training, the gradient doesn't seem to have any significant overlap in the top or next subspaces ($\sum_{i=1}^{20} c_i^2 = .20$, after 0 steps of training cf.

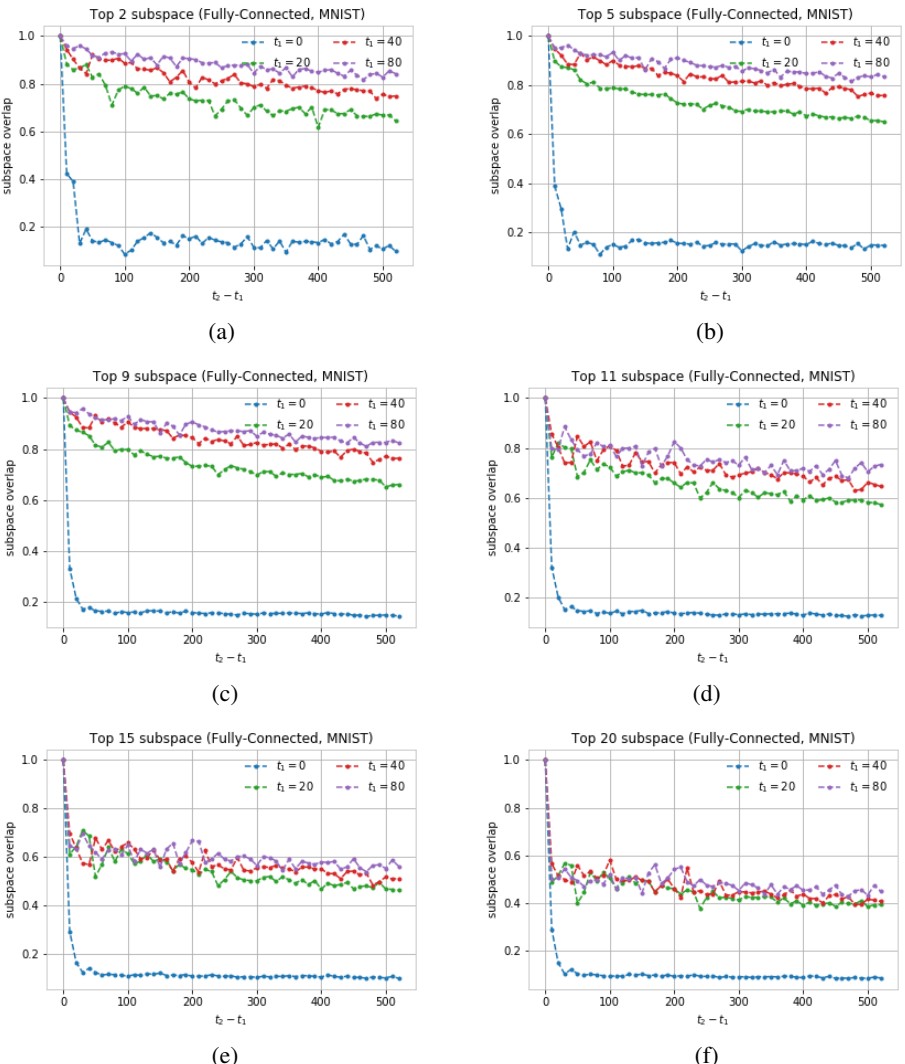

Figure 9: Overlap of top Hessian subspaces $V_{\text{top}}^{(t_1)}$ and $V_{\text{top}}^{(t_2)}$ for fully-connected network trained on MNIST using the same architecture in Figure 1. (a) Top 2 subspace. (b) Top 5 subspace. (c) Top 9 subspace. (d) Top 11 subspace. (e) Top 15 subspace. (f) Top 20 subspace.

$\sum_{i=1}^{20} c_i^2 = .94$ after 50 steps of training). After some training, see (b), (c), (d), the gradient is spread over the different $c_i^2$'s from $i = 1 \ldots 10$ in the top subspace and never has any real significant weight for $i > 10$. (E.g. we have $\sum_{i=1}^{10} c_i^2 = .93$ vs. $\sum_{i=11}^{20} c_i^2 = .01$ after 50 steps of training.)

## D ADDITIONAL EXPERIMENTS

In this section, we provide some plots highlighting additional experiments. The results of these experiments were summarized in Table 1, but we include some additional full results on the gradient overlap with the top-$k$ subspace here.

In particular, Figure 12 plots the fraction of the gradient lying in the top subspace, $f_{\text{top}}$, for a variety of different scenarios. In (a) we give an example of changing the learning rate, in (b) we give an example of changing the batch size, in (c) we give an example with 0 hidden layers, in (d) we give an example of changing the activation function, in (e) we apply a random permutation to labels, and in

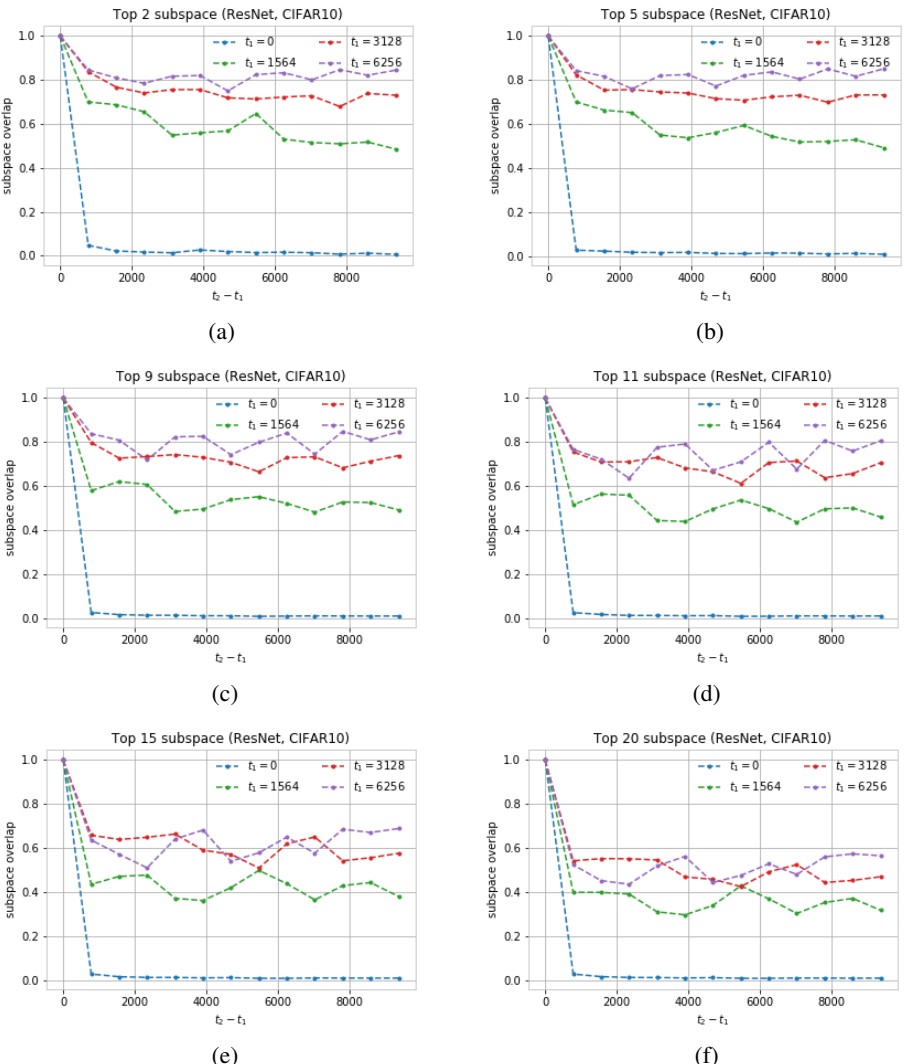

Figure 10: Overlap of top Hessian subspaces $V_{\text{top}}^{(t_1)}$ and $V_{\text{top}}^{(t_2)}$ for ResNet-18 architecture trained on CIFAR10 as in in Figure 1. (a) Top 2 subspace. (b) Top 5 subspace. (c) Top 9 subspace. (d) Top 11 subspace. (e) Top 15 subspace. (f) Top 20 subspace.

(f) we use the Adam optimizer instead of SGD. In all these experiments, we see pretty consistently that the gradient quickly converges to live in the top subspace and then stays there.

## E    ANALYTIC EXAMPLE: DETAILED CALCULATIONS

For the reduced case of a 2-sample, 2-class problem learned using softmax-regression, the loss function can be written as

$$L(\theta) = \frac{1}{2} \log \left( 1 + e^{(\theta_2 - \theta_1) \cdot \mu_1} \right) + \frac{1}{2} \log \left( 1 + e^{(\theta_1 - \theta_2) \cdot \mu_2} \right) . \tag{13}$$

At a late stage of training the loss is near its zero minimum value. The exponents in equation 13 must then be small, so we can approximate

$$L(\theta) \approx \frac{1}{2} e^{(\theta_2 - \theta_1) \cdot \mu_1} + \frac{1}{2} e^{(\theta_1 - \theta_2) \cdot \mu_2} . \tag{14}$$

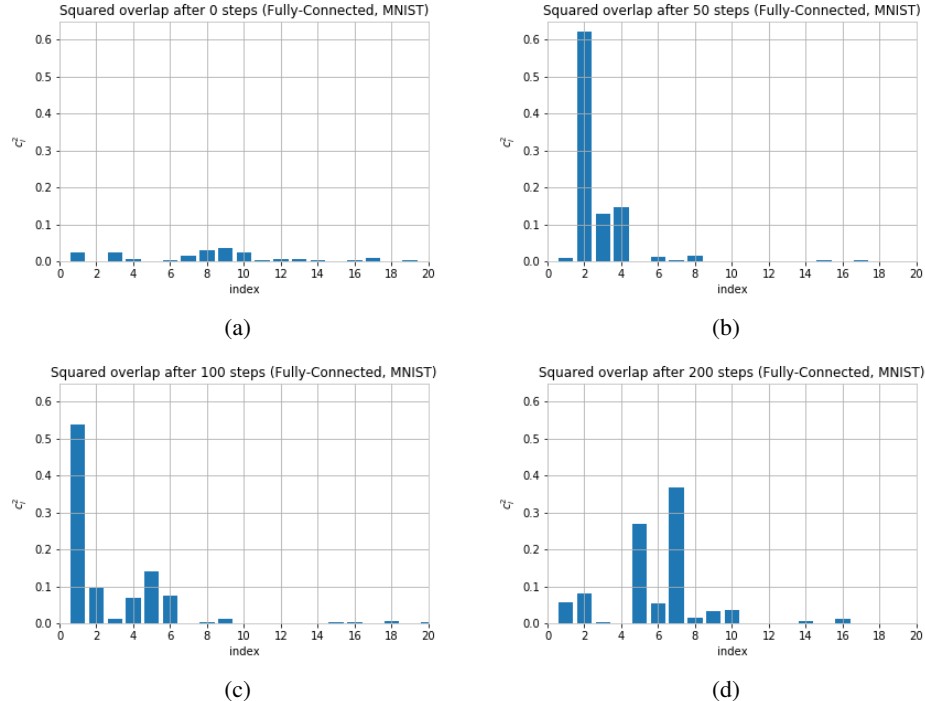

Figure 11: The overlap squared $c_i^2$ of the gradient with the $i$th eigenvector of the Hessian. Data is for a fully-connected (100,100) architecture trained on MNIST for (a) 0 steps, (b) 50 steps, (c) 100 steps, and (d) 200 steps. After 0 steps, we have $\sum_{i=1}^{20} c_i^2 = .20$ compared with $\sum_{i=1}^{20} c_i^2 = .94$ after 50 steps. Also, note that after 50 steps we have $\sum_{i=1}^{10} c_i^2 = .93$ vs. $\sum_{i=11}^{20} c_i^2 = .01$. Together, these results show that that the gradient dynamically evolves to lie mostly in the top subspace and is not simply an eigenvector of the Hessian.

The loss function has $2d - 2$ flat directions,[17] and so the Hessian can have rank at most 2, and the gradient will live inside this non-trivial eigenspace. This is a simple example of the general phenomenon we observed. To gain further understanding, we solve for the optimization trajectory.

We train the model using gradient descent, and take the small learning rate limit (continuous time limit) in which the parameters $\theta(t)$ evolve as $\frac{d\theta}{dt} = -\eta \nabla L(\theta(t))$. The general solution of this equation is

$$\theta_1(t) = \tilde{\theta}_1 + \frac{\mu_1}{2} \log\left(\eta t + c_1\right) - \frac{\mu_2}{2} \log\left(\eta t + c_2\right) , \tag{15}$$

$$\theta_2(t) = \tilde{\theta}_2 - \frac{\mu_1}{2} \log\left(\eta t + c_1\right) + \frac{\mu_2}{2} \log\left(\eta t + c_2\right) . \tag{16}$$

The space of solutions has $2d - 2$ dimensions and is parameterized by the positive constants $c_{1,2}$ and by $\tilde{\theta}_{1,2}$, which are constant vectors in $\mathbb{R}^d$ orthogonal to both $\mu_1$ and $\mu_2$. The gradient along the optimization trajectory is then given by

$$\nabla_{\theta_1} L(t) = -\nabla_{\theta_2} L(t) = -\frac{\mu_1}{2(\eta t + c_1)} + \frac{\mu_2}{2(\eta t + c_2)} = \frac{2(\mu_2 - \mu_1)}{\eta t} + \mathcal{O}(t^{-2}) . \tag{17}$$

Notice that in the limit $t \to \infty$ the gradient approaches a vector that is independent of the solution parameters.

Next, consider the Hessian. By looking at the loss equation 13 we see there are $2d - 2$ flat directions and $2d$ parameters, implying that the Hessian has at most rank 2. Let us work out its spectrum in

---

[17] There are $d$ directions spanned by $\theta_1 + \theta_2$, and $d - 2$ directions spanned by directions of $\theta_1 - \theta_2$ that are orthogonal to $\mu_1, \mu_2$.

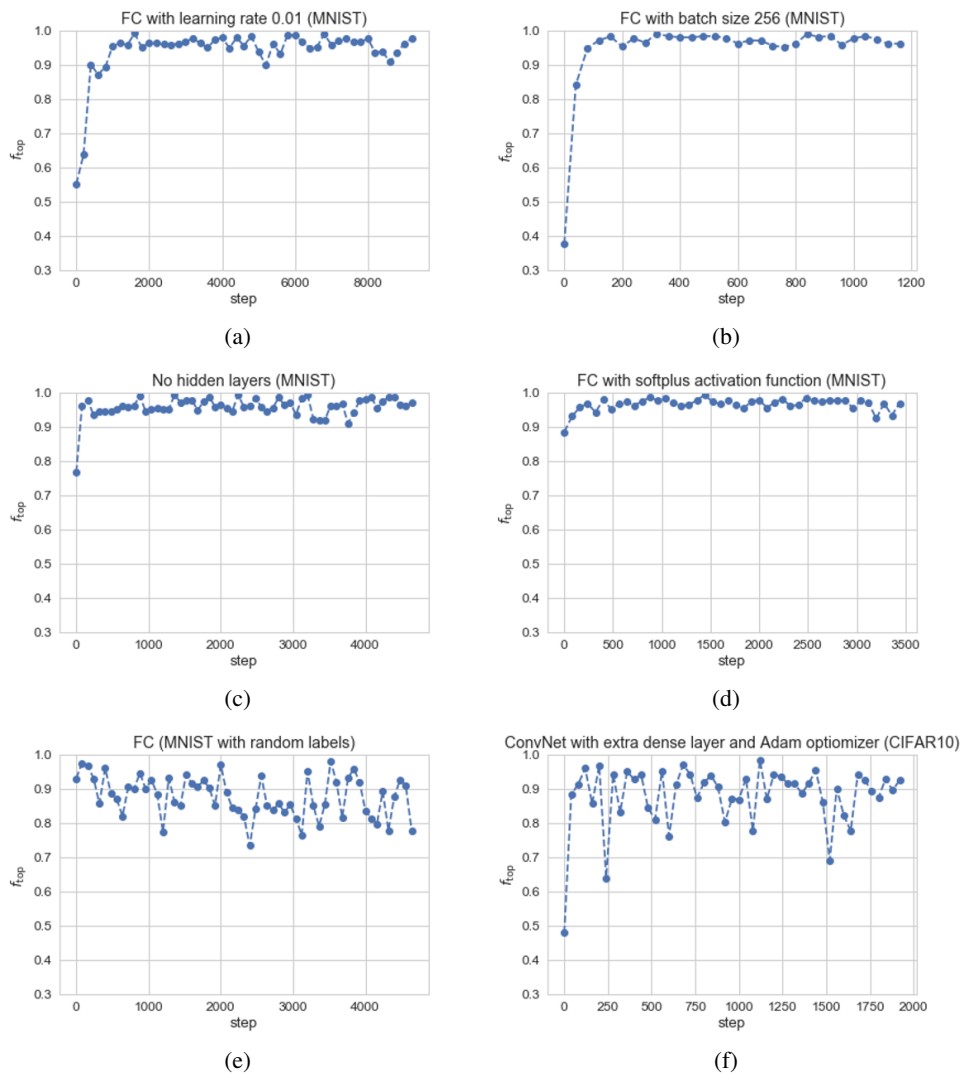

Figure 12: Fraction of the gradient in the top subspace $f_{\text{top}}$. In experiments (a)-(e), we use a fully-connected network trained on MNIST, and in (f) we use a CovNet trained on CIFAR10. The changes from the setup described in Figure 1 are: (a) changed learning rate, $\eta = .01$ instead of $\eta = 0.1$. (b) changed batch size, $256$ instead of $64$. (c) no hidden layers, just softmax. (d) changed activation: softplus instead of ReLU. (e) random labels on MNIST. (f) changed optimizer, Adam instead of SGD.

more detail. Decomposing the parameter space as $\mathbb{R}^k \otimes \mathbb{R}^d$, the Hessian along the optimization trajectory is given by

$$
\begin{aligned}
H &= \begin{pmatrix} +1 & -1 \\ -1 & +1 \end{pmatrix} \otimes \left[ \frac{\mu_1 \mu_1^T}{2(\eta t + c_1)} + \frac{\mu_2 \mu_2^T}{2(\eta t + c_2)} \right] \\
&= \frac{1}{2\eta t} \begin{pmatrix} +1 & -1 \\ -1 & +1 \end{pmatrix} \otimes \left[ \mu_1 \mu_1^T + \mu_2 \mu_2^T \right] + \mathcal{O}(t^{-2}).
\end{aligned}
\tag{18}
$$

At leading order in the limit $t \to \infty$ we find two non-trivial eigenvectors, given by

$$
\begin{pmatrix} \mu_1 \\ -\mu_1 \end{pmatrix} \quad \text{and} \quad \begin{pmatrix} \mu_2 \\ -\mu_2 \end{pmatrix},
\tag{19}
$$

both with eigenvalue $(\eta t)^{-1}$. The remaining eigenvalues all vanish. The top Hessian subspace is fixed, and the gradient is contained within this space.

