# OpenReview forum: "Gradient Descent Happens in a Tiny Subspace"
_ICLR.cc/2019/Conference_

### Official Review · AnonReviewer1 · 2018-10-30
**Interesting paper with supported experiments**

**Rating:** 6
**Confidence:** 4

**Review:**

This paper shows that gradient descent mostly happens in a tiny subspace which is spanned by the top eigenvectors of the Hessian. Empirical results are shown to support the claim. This finding is interesting and provides us some insights to design more efficient optimization algorithms. Overall, this paper is interesting and easy to follow.

The experiments in section 2 do a decent job supporting the claim that gradient descent happens in a tiny subspace and the subspace is mostly preserved over long periods of training. However, I would like to add a couple more points to the discussion:

- It's not surprising that the magnitude of gradient is larger in the high curvature directions, which means that the learning would always first happen in top subspace if the learning rate is small enough. It would be interesting to tune the parameter of learning rate to see if the phenomena would occur across different learning rate (especially large learning rate).
- The argument of gradient descent happening in a tiny space is quite obvious if the Hessian has only a few large eigenvalues. Therefore, it would be interesting to discuss the spectrum of the Hessian a little bit.
- Contrary to plain gradient descent, natural gradient is able to learn low curvature directions (small eigenvalues). It would be interesting to show some experiments with natural gradient methods.

Following section 2, the authors give a toy model to further backup their claims. However, I find the example is too restrictive and may not explain why the subspace would be preserved over the training. If I understand right, the loss function of the toy model is convex and Hessian is a constant over time. For this kind of toy model, the Hessian (or equivalently the Fisher matrix) only depends on the input distribution, so it's easy to see that the Hessian would be low-rank and preserved throughout the training. However, neural networks are highly non-convex, so it's unclear to me whether the implications of the toy model would generalize. I encourage the authors to analyze more complicated models.

To summarize, I think this paper is interesting and well-written. However, it lacks convincing explanation why the subspace would preserve over the training (to me, it's more interesting than the point that gradient descent happens in tiny subspace). Anyway, it is not completely reasonable to expect all such possible discussions to take place at once.

---

> ### Author Response · Authors · 2018-11-23
> **Response to particular points**
>
> We thank the reviewer for the kind words about our paper being interesting an easy to follow. We'd also like to respond to your individual comments.
>
> - We extensively investigated different learning rates over a few orders of magnitude, finding our results to be robust. Due to space and presentation issues, only one of these results is included (as the third element in Table 1). We've also included a plot, Figure 12 (a).
>
> The behavior with respect to changing the learning rate is reasonably well captured by our Hessian toy model (equation 8). Since eta scales with time, decreasing the learning rate has the effect of increasing the magnitude of the large eigenvalues (which will decay over the course of training). (See, e.g. Figure 8.) We don't expect this to hold if the learning rate is too big (such that the model doesn't train), but seems robust as long as the model trains.
>
> Since it's clear that perhaps our way of summarizing experiments with a Table was perhaps confusing, we included Appendix D that shows the effect of a different learning rate on our results.
>
> Responses to particular points:
>
> - The spectrum of the Hessian was discussed in Appendix B, and we've expanded quite a bit on that discussion. Furthermore, in the main text when we derive our summary metric for the gradient being in the top subspace, equation (3), we discuss the typical Hessian spectrum (given results from Sagun et al. and our experiments) and explain why this can lead to the gradient living in a small subspace. However, it's not just enough to live in a small subspace independently across updates since the eigenvectors that span the top subspace might be mixing with the bulk eigenvectors. Our notion of "living in a small subspace" is meant to refer to one that is nearly preserved over the course of training. It is for this reason that we measure the overlap of subspaces at different time, finding that the top subspace is nearly preserved. (See Figures 2 and 3.)
>
> - We agree that it might be interesting to understand whether using natural gradient still leads to the same gradient and Hessian behavior. We would also like to emphasize that it's not necessarily obvious to us whether or not SGD is learning in the low-curvature directions or what role they play. For instance, it might be the case that they are a necessary part of the dynamical mechanism that creates the nearly-preserved subspace. If that were the case, for instance, then the approximate second-order algorithm we propose with equation (9) would fail due to projecting out important information.
>
> - We agree that the toy model is simple and is not the final story about why the subspace is preserved. (We have some upcoming work that studies the k-class extension. In this case, it's not true that the Hessian naturally has rank k. It starts as rank k(k-1) and then dynamically becomes rank k.)
>
> Furthermore, it is not true that the Hessian is constant over time (see, e.g. equation 18 for its time dependence), the eigenvalues decay like ~1/t, if t is the number of updates. As we commented on above, we find this analysis illuminating as it let us understand the effect of changing the learning rate (which we then confirmed extensively via experiment).
>
> We understand that neural networks are not convex, but by simulating the k-class extension of our toy model (see section 3.1) and identifying similar behavior to the deep models we studied, we think of the toy model as a helpful illustrative example, and not the definitive explanation. We have tried to add text to make it clearer how we regard the toy model.
>
> -Finally, we agree that the preservation of the subspace is perhaps the most interesting part of this work. We have added an additional appendix (Appendix C) exploring this phenomenon in more detail, summarizing these results in Figure 3. We hope that the reviewer will appreciate these results in particular.

---

### Official Review · AnonReviewer2 · 2018-10-31
**Agreed with small subspace but not with number of classes. Needs a more thorough study.**

**Rating:** 4
**Confidence:** 4

**Review:**

The authors build on recent works that study the spectrum of the Hessian of deep networks (e.g. Sagun et al). Previous work argues that Hessian is approximately low-rank i.e. there are few large eigenvalues and many small eigenvalues. This work argues that after some training, large eigenvectors of the Hessian converges to a subspace and stays there.

Intuitively, this papers message makes sense and is interesting. I also agree with the tiny subspace argument in the paper. However, I am not convinced by a couple of things and I believe further evidence is necessary.

1) Authors claim the top subspace has same rank k (where k=number of classes) and backs this up with linear classifier with 2 class (toy model). It is clear that any noiseless k-class linear classifier has its gradient lie on a k dimensional subspace. Similarly, for a deep network, I agree with hessian of the final layer will be rank k however earlier layers can have different ranks as a function of complexity of the lower level representations. My worry is that perhaps the Hessian of the final layer is somehow dominating over the other ones. Ideally, I would like to see analysis of individual layers to reach a conclusion. Is first layer also approximately rank k?

2) Similarly, we need to see the eigenvalue distribution of the Hessian. Say there is a single very large eigenvalue and all others are small. The same claim would still hold. So it is not clear if number of classes really plays a role from the available experiments. This can indeed happen is the classes are correlated for instance. Authors can perhaps plot gradient over top k/2 subspace to reveal if their claim is specific to number of classes.

---

> ### Author Response · Authors · 2018-11-23
> **Revised with much more evidence for number of classes**
>
> We very much appreciate that you found our paper interesting and agree with our general arguments. We thank you for your very constructive feedback. Let us address your specific concerns below.
>
> 1) First, we respectfully disagree that any noiseless k-class linear classifier has its gradient lie on a k-dimensional subspace. This can be verified in a more complicated toy model of a k-class classifier where the input is a mixture of k Gaussians. In the noiseless limit, there are k(k-1) nonzero eigenvalues, and it is only through complicatedly solving the dynamics that one can see how k(k-1) relaxes to k. (This result is outside the scope of this paper and is being written up as a follow up.) However, we have added some text referring to this future work (footnote 14) and also explain this point in the current paper. (Also note that k=2 is degenerate, and so one needs to analyze higher k to see this effect. Nevertheless, we still feel that the toy model is illustrative of the behavior of the parameters, gradient, and Hessian eigenvalues over training and helps one reason about thing---e.g. the expected behavior when changing learning rate---which we later confirm on deep models via experiment.)
>
> For this reason, it's also not obvious that the final layer should only have rank k, and not rank k(k-1). (We should further note that you can study the k-class model with noise, and the noise lifts the rest of the Hessian eigenvalues to be nonzero.)
>
> On the other hand, we agree that the Hessian of the final layer might dominate over the earlier ones. If the final layer is rank k, then the equations of back propagation mean that the gradient gets projected onto this lower rank subspace, regardless of the action of the other tensors. However, we think that this is an important phenomenon that deserves to be highlighted, since it controls very directly how the gradient can update the model.
>
> 2) We very much agree that the eigenvalue distribution of the Hessian is important, and this was unfortunately relegated to Appendix B for space concerns (and since it was already explored in e.g. Sagun et al.). We have greatly expanded the discussion in this appendix. In particular, Appendix B shows an example eigenvalue distribution that highlights the k eigenvalue effect (Figure 4) and also a density plot of the top 1000 eigenvalues for CIFAR100 (Figure 5). The density plot is a histogram of log eigenvalue averaged over 200 realizations. It makes it very clear that there's some kind of change in the function describing the density distribution very near the mean 100th eigenvalue. In fact, this portion is fit very well by a Gaussian, which make it very suggestive to say that perhaps that the top-k distribution is log-normal---however far more evidence and thought would be needed in order to make this claim. In order to make our spectrum results clearer to readers, we have better highlighted Appendix B in the main text, and we have also expanded our discussion of this spectrum there.
>
> The suggestion to plot the gradient over other size subspaces is a very good one, and we thank the reviewer for encouraging us to included these results. We have a very nice summary of these results in Figure 3, and we've written a new Appendix (labeled C, with the previous Appendix C bumped up to Appendix E) that's explicitly dedicated to this question. In particular Figures 9 and 10 contain the raw data summarized by Figure 3, showing that an important change happens at a subspace dimension that's one less than the number of classes in the dataset. In Figure 11, we show that the gradient overlaps across many of the different eigenvectors in the top subspace after some training, and that the gradient doesn't really extend to the next subspace. (Figure 1 also captures this point.)
>
> We hope the reviewer finds these results to be as convincing as we do and that we have satisfied the criteria for being thorough.

---

### Official Review · AnonReviewer3 · 2018-11-04
**This paper describes an interesting phenomenon, but some of the experimental evidence is a bit lacking.**

**Rating:** 4
**Confidence:** 3

**Review:**

This paper describes an interesting phenomenon: that most of the learning happens in a small subspace. However, the experimental evidence presented in this paper is a bit lacking. The authors also cook up a toy example on which gradient descent exhibits similar behavior. Here are a few detailed comments:

1. The Hessian overlap metric is suitable for showing the gradient lies in an invariant subspace of the Hessian, but does not show it lies in the dominant invariant subspace.
2. There are well-established notions of distances between subspaces in linear algebra, and I suggest the authors comment on the connection between their notion of overlap between subspaces and these established notions.
3. The authors make a few statements along the lines of ``the Hessian is small, so the objective is flat''. This is a bit misleading as it is possible for the gradient to be large but the Hessian to be small.

---

> ### Author Response · Authors · 2018-11-23
> **Revised with a large amount of additional experimental evidence**
>
> Thank you for your specific comments. We will address them here directly.
>
> 1. It is true that having a large overlap does not immediately imply that the gradient lives in the top subspace. However, in all examples we checked, whenever the overlap was large the gradient did in fact live almost entirely in the top subspace. In the revised version we provided further empirical evidence of this relation, see Figure 11 in particular.
>
> 2. Our measure of subspace overlap, equation (5), is a natural measure of overlap. Each subspace is defined by a projector onto that subspace. The Frobenius norm of the difference ||P-P'|| is perhaps the best known measure of distance between two subspaces (and is the natural metric on the Grassmanian, the manifold of linear subspaces ). Our equation (5) is simply related to this measure (analogous to the cosine distance for vectors, and literally the cosine distance for one dimensional vectors.), ||P-P'||_2 = 2k - 2Tr(P.P'). (We've updated the draft to make this clearer.)
>
> We are interested in the overlap (Tr(P.P')/k) not the distance||P-P'|| since it measures the average fraction of a late-time vector that lives in the early-time subspace. Thus we decided on equation (5) as a very natural measure that also has this intuitive interpretation. As mentioned, we have clarified the relation to the distance measure ||P-P'|| in the text.
>
> 3. When we use the term flat, we mean in terms of curvature. A direction where the Hessian vanishes but there is a nontrivial gradient is still flat, even though the parameters update in that direction. The point is that the amount the parameters will update will be constant. However, we appreciate that this may be confusing to readers, so we have added a clarifying footnote, footnote 1, regarding this point.

---

### Author Response · Authors · 2018-11-23
**Summary of revisions and responses**

We are grateful to the reviewers for their thoughtful comments. We have tried to address their concerns directly below. We have also updated the draft (mostly by vastly expanding the Appendices) to include additional analysis, experiments, and hopefully clarity.

In particular, we've including the following important experiment evidence addressing the specific concerns of the reviewers:

-Figure 3 clearly shows that the relevant dimension for the subspace freezing is related to the number of classes in the dataset. Appendix C is a more in depth discussion of this, with additional experimental evidence.
-Appendix B includes an extended discussion of the Hessian spectrum. Let us highlight some important plots. Figure 4 is an example spectrum for a single realization, with the eigenvalues in the top subspace now labeled. Figure 5 shows a realistic model on a dataset with a different number of classes than MNIST and CIFAR10 (CIFAR100) averaged over 200 realizations. The feature around the mean 100th eigenvalue is clear.
-Figures 6 and 7 plot a particular eigenvector at different points in training, showing that such eigenvectors both evolve and are not dominated by any particular parameter or layer.
-Figure 11 makes it clear that the gradient is not actually an eigenvector of the Hessian as some reviewers were concerned about. It also makes it clear that the gradient overlaps are nicely spread over the top subspace and evolving nontrivially in time.
-Appendix D contains additional experimental results for different hyperparameters, including showing robustness to changing the learning rate as Reviewer 1 requested.
-We've tried to add clarifying comments to the draft about the utility of the toy model, see e.g. footnote 14 in particular.
-We also want to highlight that Figure 2 (as well as the new Figure 3 and Figure 10) contains results for the freezing of the subspace for the ResNet 18 and are consistent with the claims of the paper.

As a general comment, Reviewers 2 and 3 were concerned that the experimental evidence is lacking for our observed phenomenon. We hope this additions will help put those concerns to rest. We hadn't included these results before because we didn't appreciate that we can expand our Appendices in this way. Naturally, in the experimental phase of the project, we had conducted many more experiments than could possibly fit in the paper. For clarity, we had thought to focus on three illustrative examples, a fully-connected network, a simple convolutional network, and a ResNet-18. Rather than clutter the paper with an overabundance of plots, we developed a summary metric for our phenomenon, the overlap described in equation (3) and summarized a variety of additional experiments (which still do not cover the full number of experiments we performed). Of course when performing these additional experiments, we checked that the effect illustrated directly in Figure 1 also covers these other scenarios, and we offer the explanation around equation (4) as to why our summary metric (3) is a reasonable metric. Nevertheless, as we stated above, we created Appendix D with the fractional of gradient in top subspace plots for many of the experiments collected in Table 1. We hope these results will be useful in considering the robustness of the phenomena we discuss.

Furthermore, all three reviewers found our toy model unsatisfying. We would like to highlight that we interpret the toy model to be illustrative (and was helpful in understanding how our results might change if we change hyperparameters such as learning rate) and not a definitive explanation. We find it helpful to have a model we can solve that also illustrates many of the phenomenon we see in the more complicated models. However, we have added text to the body making it clear that we don't think of the toy model as a complete explanation. Finally, we have added a reference (footnote 14) that talks about upcoming work, the k-class extension, giving one part of the result that explains why having k eigenvalues (instead of k(k-1) as should be naively expected by symmetries) is a nontrivial result.

We hope that these revisions are helpful in making our paper clearer and will convince the reviewers that our results are robust across many experiments. Please feel free to request additional clarifications or explanations.

---

### Public Comment · ~Nicolas_Le_Roux2 · 2018-12-14
**Link with "MEASURING THE INTRINSIC DIMENSION OF OBJECTIVE LANDSCAPES"**

Hi,

I am trying to reconcile these findings with those of https://arxiv.org/pdf/1804.08838.pdf where they find that the intrinsic dimensionality had more to do with the input dimension than the number of classes.

One of the differences between the two works is that your subspace is not randomly chosen but I was wondering if you had insights on the discrepancy between the two claims.

---

### Meta-Review · Area_Chair1 · 2018-12-13
**Summary review**

**Confidence:** 5
**Recommendation:** Reject

**Metareview:**

The paper is overally interesting and addresses an important problem, however reviewers ask for more rigorous empirical study and less restrictive settings.